# Conjugation Operons in Gram-Positive Bacteria with and without Antitermination Systems

**DOI:** 10.3390/microorganisms10030587

**Published:** 2022-03-08

**Authors:** Andrés Miguel-Arribas, Ling Juan Wu, Claudia Michaelis, Ken-ichi Yoshida, Elisabeth Grohmann, Wilfried J. J. Meijer

**Affiliations:** 1Centro de Biología Molecular Severo Ochoa (CSIC-UAM), Instituto de Biología Molecular Eladio Viñuela (CSIC), C. Nicolás Cabrera 1, Universidad Autónoma, Canto Blanco, 28049 Madrid, Spain; amiguel@cbm.csic.es; 2Centre for Bacterial Cell Biology, Medical Faculty, Biosciences Institute, Newcastle University, Richardson Road, Newcastle upon Tyne NE2 4AX, UK; l.j.wu@newcastle.ac.uk; 3School of Life Sciences and Technology, Berlin University of Applied Sciences, Seestrasse 64, 13347 Berlin, Germany; claudia.michaelis@bht-berlin.de; 4Department of Science, Technology and Innovation, Kobe University, 1-1 Rokkodai, Nada, Kobe 657-8501, Japan; kenyoshi@kobe-u.ac.jp

**Keywords:** conjugation, antibiotic resistance, Gram-positive bacteria, antitermination, pLS20, pIP501, *Bacillus subtilis*, *Enterococcus faecalis*

## Abstract

Genes involved in the same cellular process are often clustered together in an operon whose expression is controlled by an upstream promoter. Generally, the activity of the promoter is strictly controlled. However, spurious transcription undermines this strict regulation, particularly affecting large operons. The negative effects of spurious transcription can be mitigated by the presence of multiple terminators inside the operon, in combination with an antitermination system. Antitermination systems modify the transcription elongation complexes and enable them to bypass terminators. Bacterial conjugation is the process by which a conjugative DNA element is transferred from a donor to a recipient cell. Conjugation involves many genes that are mostly organized in one or a few large operons. It has recently been shown that many conjugation operons present on plasmids replicating in Gram-positive bacteria possess a bipartite antitermination system that allows not only many terminators inside the conjugation operon to be bypassed, but also the differential expression of a subset of genes. Here, we show that some conjugation operons on plasmids belonging to the Inc18 family of Gram-positive broad host-range plasmids do not possess an antitermination system, suggesting that the absence of an antitermination system may have advantages. The possible (dis)advantages of conjugation operons possessing (or not) an antitermination system are discussed.

## 1. Introduction

Conjugation is a horizontal gene transfer (HGT) process by which a conjugative element (CE) is transferred from a donor to a recipient cell through a channel connecting the two cells. CEs can be integrated in bacterial chromosomes, which are named integrative and conjugative elements, or on plasmids, named conjugative plasmids (for review, see [1,2,3]). CEs often carry antibiotic resistance (AR), toxin and/or virulence genes [4]. Conjugation is the main HGT route responsible for the distribution of these pernicious genes [5]. AR, in particular, causes much damage, being responsible for tens of thousands of human deaths annually as well as large economic losses [6]. In addition to CEs, some Gram-positive (G+) bacteria also harbor small plasmids that encode AR genes and a relaxase gene, which allows the plasmid to generate a single-stranded DNA. When present alone in a cell, these small plasmids are unable to transfer horizontally to other cells. However, in the presence of a conjugative plasmid, the ssDNA can be transported into another cell by exploiting the connecting channel generated by the co-residing conjugative plasmid in a process named mobilization [7,8]. Thus, CEs can horizontally spread not only AR and other pernicious genes located on the CEs, but also those on mobilizable plasmids via mobilization. A detailed understanding of the different conjugation steps is a prerequisite to developing strategies or drugs that can impede the conjugation-mediated spread of AR and other pernicious genes. 

CEs contain all of the genes necessary for carrying out the four principal steps of conjugation: firstly, the recipient cell selection and attachment; secondly, the generation of the connecting channel that is a Type IV secretion system; thirdly, the processing of the DNA to generate an ssDNA copy of the CE, which in almost all conjugative systems is the DNA form that is transferred; and finally, transport of the ssDNA through the channel into the recipient cell, and subsequent conversion of the ss into double-stranded DNA. So far, most conjugation studies are based on CEs from Gram-negative (G-) bacteria. Among the best-studied CEs of G+ bacteria are the conjugative plasmids pIP501 from *Streptococcus agalactiae*, pCF10 from *Enterococcus faecalis*, pCW3 from *Clostridium perfringens* and pLS20 from *Bacillus subtilis* [9,10,11,12,13,14,15,16]. 

The expression of the multiple genes involved in the conjugation process poses a high energetic burden on the cell. In addition, conjugation has large impacts on the host cell and on the plasmid itself; for example, the conjugation proteins alter the surface and membrane characteristics of the host, and the replication mode of the plasmid changes from the theta to the rolling circle type of replication to generate the ssDNA form that is transferred. These are probably the reasons why, in most cases, the conjugation genes are clustered together in one operon that is preceded by a promoter whose activity is strictly controlled [11,15,16,17]. However, this organization also has disadvantages. For instance, it is very likely that proper functioning of the conjugation process requires (i) different levels of proteins encoded by the different genes, and (ii) temporal regulation of expression of proteins involved in the different stages. In part, temporal expression may be achieved by the order of genes within an operon, and the level of proteins can be modulated by translation efficiency and mRNA stability. However, these fluctuations are limited when compared to the wide range in expression levels that can be achieved by placing genes under the control of promoters with different strengths. Another drawback of organizing genes in operon structures, especially large ones, is spurious transcription, which is the generation of unintended RNA transcripts due to transcription initiation events at non-promoter sites (aka cryptic promoters) [18,19,20]. Such spurious transcription occurs on a large scale in bacteria [21]. In the case of operons, it undermines the strict control of the main promoter, resulting in the undesired expression of some or many of the genes in the operon. 

Bacteria possess two types of termination signals: intrinsic and Rho-dependent terminators [22,23,24,25,26,27]. In G+ bacteria, intrinsic terminators are mainly used as termination signals for coding DNA sequences. Intrinsic terminators are typically characterized by a GC-rich inverted repeat separated by a few base pairs and followed by a stretch enriched in Ts in the non-template DNA strand. When transcribed into RNA, this region forms a hairpin structure followed by a U-rich tract, which is sufficient to terminate transcription [28,29].

Recently, we reported that pLS20 is the prototype of a family of related plasmids and that the conjugation operons present on all these plasmids (i) contain multiple intrinsic terminators and (ii) start with a processive antitermination (P-AT) system, named *conAn* [30,31]. The *conAn* system contributes to the strictly controlled expression of the conjugation genes by minimizing the effects of spurious transcription and allowing the differential expression of subsets of genes within the conjugation operon. P-AT systems function by altering the transcription elongation complexes (TECs) through interaction with an antiterminator factor, allowing the altered TECs to read through (multiple) transcription terminating signals (for review, see [32,33]). In most cases, the antiterminator factor is a protein. The best-studied P-AT systems are based on the antiterminator proteins N and Q of the *Escherichia coli* phage lambda [34,35]. Other protein-based P-AT systems concern analogues of the transcription elongation factor NusG [36]. Two systems have also been described in which the transcription factor was not a protein, but an RNA: the *put* system of *E. coli* phage HK022 (a λ phage) and the EAR systems present in the exopolysaccharide operons of *B. subtilis* and other bacilli [37,38,39]. The *conAn* system present at the start of the conjugation operon on plasmid pLS20 and related plasmids is different because it is composed of two components: ConAn1 (protein) and *conAn2* (RNA). Whereas *conAn2* is responsible for antitermination, ConAn1 acts as a processivity factor allowing antitermination to take place even at large distances from the promoter. Only transcription elongation complexes derived from the main conjugation promoter of pLS20, named P*_c_*, can become loaded with ConAn1 and *conAn2.* The altered complexes are then able to read through the >20 terminators inside the pLS20 conjugation operon [30]. Any transcription elongation complexes derived from spurious transcription events will not be associated with the *conAn1* and *conAn2* components, and so will stop at the first terminator encountered. The *conAn* system also allows differential expression of a subset of genes. One of the examples is the third gene of the pLS20 conjugation operon, which is preceded by a constitutive weak promoter. Transcripts starting at this promoter end at a terminator located two genes downstream, resulting in the constitutive low-level expression of these two genes. However, upon activation of the much stronger conjugation promoter P_c_, these two genes become highly expressed. One of these two genes encodes the surface exclusion protein, and the differential expression of this protein has important consequences for the functionality of the exclusion system [40]. Thus, the presence of a *conAn* type P-AT system seems to be beneficial by allowing the differential expression of subsets of genes within the conjugation operon, while contributing to strict regulation of the conjugation genes. This may suggest that all conjugation operons in G+ bacteria are furnished with a *conAn* or perhaps other type of P-AT system. 

The Incompatibility 18 (Inc18) group of plasmids contains one or more antibiotic resistance genes encoding resistance to vancomycin, chloramphenicol and the macrolide–lincosamide–streptogramin (MLS) group of antibiotics. These plasmids have a broad host range and have frequently been found in bacterial genera, causing nosocomial infections such as enterococci and staphylococci [41,42,43]. The DNA replication and segregation modules of the Inc18 plasmids are conserved and share > 92% identity at the DNA level [44]. These essential modules can be combined with additional non-essential modules such as AR genes. Only a subgroup of the Inc18 plasmids contains a conjugation module allowing conjugative transfer [44]. Plasmids of this Inc18 subgroup have been shown to be responsible for vancomycin resistance transfer to *Staphylococcus aureus*. Most vancomycin resistant *S. aureus* (VRSA) are MRSA isolates that have acquired *vanA*-mediated vancomycin resistance from enterococci [45] via a pSK41-like staphylococcal conjugative plasmid, most likely pWZ909 [46,47]. The best-studied conjugative plasmids of the Inc18 group are pIP501 (30.6 kb), pAMβ1 (27.8 kb) and pRE25 (50.2 kb) [12,43,48,49]. Here, we demonstrate that plasmid pIP501, and probably also pAMβ1 and pRE25, do not contain a P-AT system. This suggests that, besides the advantages, the presence of a P-AT system also attributes disadvantages. The possible (dis)advantages of a P-AT system and the implications for host range are discussed. 

## 2. Materials and Methods

### 2.1. Bacterial Strains, Plasmids, Media and Oligonucleotides 

*B. subtilis* and *E. coli* strains were grown in lysogeny broth (LB) [50], without added glucose. All bacteria were grown in liquid media with shaking or on 1.5% LB agar plates at 37 °C. When appropriate, media were supplemented with the following antibiotics: ampicillin (Amp, 100 μg/mL) for *E. coli*, and spectinomycin (Spec, 100 μg/mL) for *B. subtilis*. The *B. subtilis* strains used were isogenic with *B. subtilis* strain 168. The bacterial strains, plasmids and oligonucleotides are listed in Table 1. All oligonucleotides were purchased from Integrated DNA Technologies (IDT) (Leuven, Belgium).

### 2.2. Construction of Plasmids and Strains

DNA techniques were performed using standard molecular methods [51]. The plasmid isolation from *E. coli* and PCR fragment purification were performed using “Wizard Plus SV Minipreps DNA Purification Systems” and “Wizard SV Gel and PCR Clean-Up System” (Promega), respectively. Plasmid pEEF7 was constructed as follows. First, complementary primers containing the Ter1 sequence were annealed in a 200 μl reaction mixture containing 2000 pmol of each primer (in 50 mM NaCl, 40 mM Tris pH 7.5), by boiling for 10 min and then slowly cooling down to room temperature. Next, the hybridized oligonucleotides were used as the insert and were ligated to plasmid pAND101 digested with *Sal*I and *Nhe*I restriction enzymes. All enzymes used were purchased from New England Biolabs, USA. The sequences of the cloned fragments were verified by DNA sequencing.

### 2.3. Transformation

Chemically competent *E. coli* cells were prepared by the fermentation service of Centro de Biología Molecular Severo Ochoa (CBMSO) and transformation was carried out using standard methods [51]. The generation of naturally competent *B. subtilis* cells and transformation were performed as described [52]. Plasmid pEEF7 was used to transform *B. subtilis* 168 competent cells and transformants were selected on LB plates supplemented with spectinomycin. Transformants resulting from double cross-over events were identified by amylase-negative phenotype. 

### 2.4. Flow Cytometry

Fluorescence quantification using flow cytometry was performed as described before [40]. In short, overnight cultures (37 °C) of strains containing a transcriptional *gfp* fusion were diluted 100-fold in prewarmed LB medium and grown at 37 °C with shaking (180 rpm) until the cultures reached an OD_600_ of 0.8–1. The cells were then collected by centrifugation (1 min 14,000 rpm) in 2 mL Eppendorf tubes. After two washing steps with phosphate buffered saline (PBS, 146 mM NaCl, 2.7 mM KCl, 8 mM Na_2_HPO_4_, and 1.5 mM KH_2_PO_4_, pH 7.0) solution passed through a 0.22 μm filter (Merck Millipore, Burlington, MA, USA) to remove any small particles, the cells were resuspended in 2 mL of filtered PBS. The fluorescence levels were expressed as the mean value of the Geomean values of 100,000 cells obtained in three independent experiments.

### 2.5. Identification of Putative Termination Sequences

The DNA sequences of conjugative plasmids were screened for the presence of putative Rho-independent transcriptional terminators by (i) the “ARNold” Web server (rna.igmors.u-psud.fr/toolbox) that uses two algorithms: Erpin and RNAmotif [53,54], and by (ii) the TransTermHP Web server (transterm.cbcg.umd.edu), which uses an algorithm that is distinct from the Erpin and RNAmotif algorithms [55].

## 3. Results

### 3.1. Highly Similar Conjugation Operons of Inc18 Plasmids pIP501, pRE25 and pAMβ1 Contain Two Putative Transcriptional Terminators 

Plasmids in a subgroup of the Inc18 family contain a conjugation operon and are self-transmissible [43,44], of which pIP501, pAMβ1 and pRE25 are the best-studied [12,43,48,49]. These conjugation operons are generally smaller than those of the pLS20-family plasmids, which all contain a P-AT system. We wanted to know whether these conjugation operons also contain a P-AT system. Interestingly, in silico analyses did not result in the identification of sequences showing similarity to the *conAn* systems or other antitermination systems. We then performed in silico analyses to investigate whether the conjugation operons of these three plasmids contain putative transcriptional terminator(s), because the presence of intrinsic transcriptional terminators within an operon is a good indication that the operon contains a P-AT system. First, we analyzed the similarity between these conjugation operons, which revealed that their DNA sequences are highly conserved (>97%) (Appendix A). Next, we screened the sequences of the three conjugation operons for the presence of putative transcriptional terminators. For each conjugation operon, two possible intrinsic terminators were detected, which we tentatively named Ter1 and Ter2. These terminators are located at identical positions in the three conjugation operons. Moreover, the sequences of the three Ter1 terminators are identical, and except for one position in the trailing T-stretch, the Ter2 sequences are also identical (Table 2).

To rule out the possibility that the slight differences in the sequences of the conjugation operons between pIP501, pAMβ1 and pRE25 might result in the presence or absence of a putative terminator, all of these regions were screened for the possibility of forming dyad symmetries preceding stretches enriched in Ts. These analyses did not reveal additional putative terminators besides Ter1 and Ter2 identified by algorithms designed to detect intrinsic terminators. 

A schematic overview of the almost 15 kb conjugation operon of pIP501, in which the positions of the predicted terminators Ter1 and Ter2 are indicated, is shown in Figure 1A.

As expected, one of the two putative intrinsic terminator signals (Ter2) is present downstream of the last conjugation gene *traO* (Figure 1A). Previous results have provided strong evidence that this is a functional terminator in vivo [56] (see Discussion). The predicted Ter2 sequences of pAMβ1 and pRE25 are also located immediately downstream of the last gene in the operon, which encode TraO homologues (Appendix A). The only other predicted terminator, Ter1, is located in the *traE* gene of pIP501, pAMβ1 and pRE25 (Figure 1A,B and Appendix A). This is in great contrast to pLS20, where 23 putative terminators were found within the conjugation operon [30]. 

### 3.2. Putative Terminator Ter1 Present in Conjugation Operons of pIP501, pAMβ1 and pRE25 Is Not Functional In Vivo 

Previously, we constructed an in vivo terminator screening system based on the *B. subtilis amyE* integration vector pAND101, which contains a *gfp* reporter gene controlled by the IPTG-inducible P*_spank_* promoter, and a multiple cloning site in between P*_spank_* and *gfp* (see Figure 2 for schematic representation). Derivatives of pAND101 were generated by cloning the DNA fragment predicted to contain a functional terminator in between the P*_spank_* promoter and the *gfp* gene. These plasmids were then used to construct isogenic *B. subtilis* strains containing a single copy of the cassette “P*_spank_*-*gfp*” or “P*_spank_*-[fragment X]-*gfp*” at their *amyE* loci. 

When grown in the presence of IPTG, the cells of the *B. subtilis* control strain AND101 lacking a terminator (P*_spank_*-*gfp*) are highly fluorescent, but isogenic cells containing a functional terminator (P*_spank_*-[Ter]-*gfp*) are not or are less fluorescent, similar to AND101 cells growing in the absence of IPTG [30]. We used this system to construct strain EEF7 (P*_spank_*-[Ter1]-*gfp*) and then used flow cytometry analysis to determine the fluorescence levels of EEF7 cells, and those of the control strains AND101 (P*_spank_*-*gfp*) and AND127 (P*_spank_*-[Ter*_30_*]-*gfp*), with the latter containing the functional terminator Ter*_30_* located downstream of pLS20 gene *30* [30]. The results of the flow cytometry experiments are shown in Figure 3. As expected, high and low fluorescence levels were obtained for the control strains AND101 (P*_spank_*-*gfp*) and AND127 (P*_spank_*-[Ter*_30_*]-*gfp*), respectively, when cells were grown in the presence of 1 mM IPTG. The fluorescence levels of EEF7 cells containing the putative terminator Ter1 were similar to those obtained for cells of the control strain without a terminator (AND101), demonstrating that the cloned fragment does not encode a functional terminator.

In summary, the only putative terminator identified within the conjugation operon of plasmids pIP501, pAMβ1 and pRE25 by in silico analysis appears not to be functional in vivo. 

### 3.3. Additional Evidence That pIP501, pAMβ1 and pRE25 Do Not Contain a Processive Antitermination System: Their Conjugation Operons Start with Relaxase Gene 

The minimization of the deleterious effects of spurious transcription will be most effective when an antitermination system is located at the start of an operon. Indeed, the *conAn*-type antitermination systems present in the conjugation operons of pLS20 and other related conjugative plasmids of G+ bacteria are all located at the start of the operon [30]. However, the main conjugation promoter, P*_tra_* of pIP501, and also pAMβ1 and pRE25, is located 140 bp upstream of *traA*, encoding the relaxase (Appendix A), and for pIP501, it has been shown that TraA regulates the activity of this promoter [11,43,56] (see Figure 4 for a schematic representation). Therefore, the genetic organization of plasmids pIP501, pAMβ1 and pRE25 is fundamentally different from that of pLS20 and the other plasmids containing a *conAn*-type antitermination system. 

## 4. Discussion

Recently, it has been shown that many conjugative plasmids of Gram-positive bacteria contain a *conAn* type P-AT system that is located at the start of the conjugation operon. Antitermination systems provide at least two benefits: they allow the differential expression of subsets of genes within the conjugation operon, and they contribute to the strict control of the expression of the conjugation genes by minimizing the deleterious effects of spurious transcription [30]. Based on this, it seemed plausible that all conjugative operons on Gram-positive plasmids contain an antitermination system. However, here we show that this is not the case. Firstly, no sequences sharing similarity with P-AT systems could be identified on the conjugation operons present on the Inc18 plasmids pIP501, pAMβ1 and pRE25. Secondly, we provided evidence that these conjugation operons do not contain any functional intrinsic terminators within the operon. Thus, although the conjugation operons of these three plasmids were predicted by in silico analysis to contain one intrinsic terminator (Ter1), the results of our terminator screening assays showed that it was not a functional terminator in vivo. Furthermore, it has been reported that the presence of three uridines immediately following the stem loop is the most highly conserved characteristic of intrinsic terminators, and that its presence is critical for terminator functionality [57,58]. Inspection of the Ter1 sequence shows that the stem of the putative Ter1 sequence is followed by two adenosines, not uridines, which could explain why this sequence does not generate a functional terminator. In the case of pIP501, our conclusion that its conjugation operon does not contain a functional intrinsic terminator is further supported by data from our previous Reverse-Transcription (RT)-PCR analyses, which showed co-transcription between flanking genes in the conjugation operon, including the genes flanking Ter1. These analyses also provided evidence that the second putative terminator that we identified, Ter2, located downstream of the last conjugation gene *traO*, is functional, because no co-transcription could be detected when a primer pair between pIP501 *traO* and the downstream gene *copR* was used in the RT-PCR experiments [56,59]. 

In addition to the absence of a functional terminator in the conjugation operon, several other lines of evidence support our view that the conjugation operons of these Inc18 plasmids do not contain an antitermination system. First, in conjugation operons containing *conAn* systems, as well as in most other operons containing a P-AT system, the gene(s) of the P-AT system are located near the start of the operon. The conjugation operons of the Inc18 plasmids we analyzed all start with the relaxase gene. 

The absence of an antitermination system in these conjugation systems prompted us to speculate whether not possessing an antitermination system could provide benefits for the plasmid that outweigh the advantages of having an antitermination system, and how the absence of an antitermination system might affect the conjugation process. Below, we discuss these possible effects and benefits. 

Spurious transcription will affect large operons more than small ones. The conjugation operon of plasmid pLS20 has a size of >30 kb [30]. Recently, we have shown that pLS20 is the prototype of a family of related conjugative plasmids that all contain a *conAn*-type antitermination system [31]. Like pLS20, the conjugation operons of the other plasmids in the family, and those present on non-pLS20 family plasmids containing a *conAn* type antitermination system, all have a size of >30 kb [30,31]. The conjugation operons on the Inc18 plasmids, on the other hand, are about half the size (about 15 kb). It is possible that the smaller size of the conjugation operons of the Inc18 plasmids is a result of the lack of an antitermination system. The size difference of the conjugation operons between the Inc18 plasmids and those containing a *conAn* antitermination system is intriguing. On the one hand, this might be an indication that the conjugation genes on the Inc18 plasmids are distributed in more than one operon, as is the case in the conjugative plasmid RP4 from Gram-negative bacteria [60]. However, there is no evidence that supports this hypothesis. On the other hand, the genes present in the conjugation operons on the Inc18 plasmids may correspond to the minimum set of genes required for conjugation, and the non-orthologous genes present in the large conjugation operons may play auxiliary roles in conjugation. The fact that the conjugation operon of pLS20 and many conjugation operons of its related plasmids contain a *rok* gene, whose encoded protein inhibits the developmental pathway of competence of its host *B. subtilis* [61], supports the latter hypothesis. Therefore, the identification of the orthologous genes present in conjugation operons of both small and large sizes may help distinguish the genes encoding essential and auxiliary roles in conjugation. 

Effective antitermination requires that the component(s) of an antitermination system makes specific interactions with constituents of the TEC of the host, enabling the adapted complex to bypass terminators, which is necessary for the proper expression of the conjugation genes and, hence, conjugation. In the case of the *conAn*-type antitermination system, we have previously shown that (i) processive antitermination requires both ConAn1 and *conAn2* to be functional in the host, and (ii) the *conAn* components encoded by different plasmids show different host-range specificities [30]. Thus, the *conAn*-type antitermination systems, and likely also other antitermination systems, will pose limitations to the host range in which the conjugation process can function. This view may be supported by the observation that plasmids of the pLS20 family are all harbored by different *Bacillus* species, suggesting that they have a narrow host range [31]. Consistent with this view, the Inc18 plasmids are known to have a very broad host range [43,62,63,64]. It has been shown that the plasmids pIP501 and pAMβ1 can be introduced by conjugation into a wide variety of Gram-positive bacteria, and that the conjugation systems of these plasmids are functional in non-native hosts [43,65,66,67,68,69]. It should be noted that the host range of a conjugative plasmid depends not only on the functionality of the conjugation system, but also on its replication functions. It is therefore important to determine whether the replication functions of pLS20 family plasmids can sustain replication in bacteria other than bacilli. 

To date, the only known P-AT system that affects the expression of conjugation operons in Gram-negative bacteria is mediated by the antiterminator RfaH, a chromosomally encoded protein previously named SfrB [70]. RfaH is a NusG paralog. The association of RfaH instead of NusG with the RNA polymerase prevents Rho-dependent termination and increases the transcription processivity of the conjugation operon (for review, see [33,71,72]). RfaH is recruited by binding to the non-template DNA strand of the operon polarity suppressor (*ops*) site (GGGCGGTAGCGT) located in the conjugation operon. All F-like plasmids, which have narrow host ranges, contain *ops* sites in their conjugation operons [72]. Interestingly, the conjugation genes in the Gram-negative broad host range plasmid RP4 are distributed in two different operons, and the largest operon is similar to the conjugation operons of the Inc18 plasmids (~15 kb) in terms of size. However, the operon contains an *ops* site, suggesting that it is also regulated by the RfaH P-AT system [73]. The RfaH-based P-AT systems are different from the *conAn* systems in several aspects: RfaH is host-encoded and functions in trans, and the system primarily acts on Rho-dependent terminators. We do not know whether the relationships between the presence of a P-AT system, the size of the conjugation operon, and the host range that we have observed also apply to Gram-negative bacteria, and whether the difference in the cell wall structure between the Gram-positive and negative bacteria influences these relationships.

In summary, here we have shown that the conjugation operons present on the plasmids of Gram-positive bacteria can be equipped with or without an antitermination system. Based on our current knowledge, the presence of an antitermination system provides benefits by allowing differential gene expression and stricter control of the expression of the conjugation genes. However, antitermination factors display different host-range compatibilities with components of the TEC of the hosts, which will limit the potential conjugative spread of the plasmid. This drawback does not apply to conjugative plasmids not containing an antitermination system, whose conjugation operon seems to be considerably smaller, permitting them to have a broader conjugative host range in return for not having the advantages that antitermination systems and bigger operons can provide. Additional research is needed to confirm this view. If verified, this will have important implications for our understanding of the conjugation-mediated spread of AR, virulence and toxin genes.

## Figures and Tables

**Figure 1 microorganisms-10-00587-f001:**
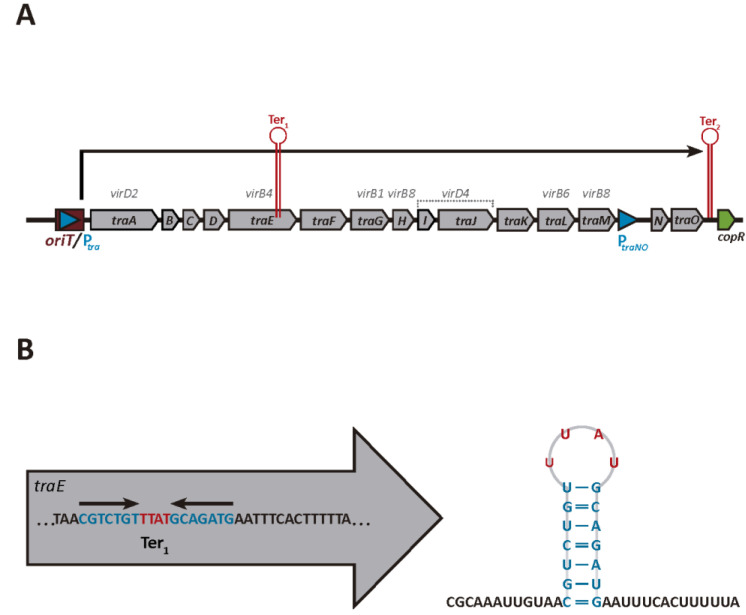
Features of the conjugation operon of plasmid pIP501. (**A**) Schematic genetic map of the pIP501 conjugation operon. Genes are indicated with arrows. The positions of the putative terminators Ter1 and Ter2 are indicated with red lollipops. *oriT* = origin of transfer. The conjugation promoter *P_tra_* is followed by *traA*, the first gene in the operon. The functions of the following *tra* genes are known: *traA* (relaxase), *traE* (ATPase), *traG* (lytic transglycosylase), *traH* (secretion channel protein), *traI* and -*J* (coupling protein), *traL* (secretion channel protein), *traM* (secretion channel protein), *traN* (repressor). (**B**) Schematic view of the putative terminator Ter1 located within the *traE* gene. Part of the *traE* sequence encompassing the putative intrinsic terminator is shown. Inverted repeat sequences and the nucleotides separating them are given in blue and red, respectively (left panel). The stem loop structure predicted to be formed when this region is transcribed into RNA is shown on the right using the same color code as in the left panel.

**Figure 2 microorganisms-10-00587-f002:**
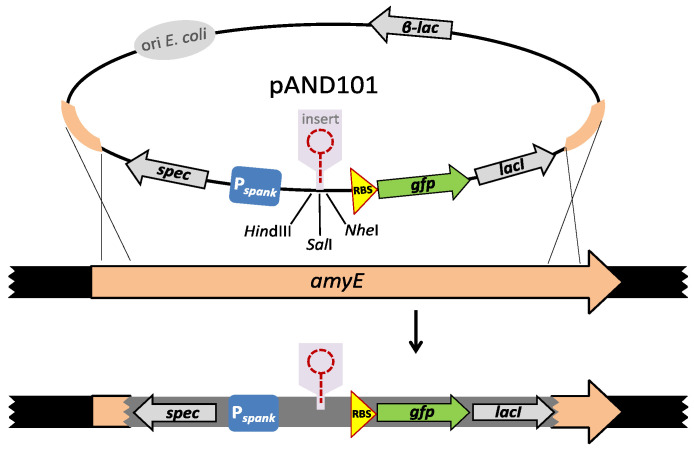
Features of the transcription terminator screening vector pAND101. Top: Vector pAND101 is a derivative of the *B. subtilis amyE* integration vector pDR110. A *colE1* origin of replication (*ori E. coli*) confers replication in *E. coli* and the β-lactamase gene (β-lac) provides resistance to ampicillin in *E. coli*. The remaining part of the vector corresponds to the cassette that becomes integrated at the *amyE* locus when the plasmid is introduced into *B. subtilis* cells by transformation. The regions corresponding to the N- and C-terminal regions of the *amyE* gene are shown in orange. Bottom: Schematic presentations of the double cross-over recombination event. The genomic configuration of the *B. subtilis amyE* region of the chromosome in the parent strain and the resulting transformants are shown below the circular plasmid. The thick black lines at the two ends indicate *B. subtilis* chromosomal DNA. The cassette that becomes integrated via double cross-over at the *amyE* locus (not to scale) encompasses the spectinomycin resistance gene (gray arrow indicated with “spec”), the *lacI* gene encoding the P*_spank_* repressor (gray arrow indicated with “*lacI*”), which is under the control of a constitutive promoter, and the *gfp* reporter gene (green arrow indicated with “*gfp*”) that is under the control of the IPTG-inducible P*_spank_* promoter (blue box indicated “P*_spank_*”). The unique restriction sites *Hin*dIII, *Sal*I and *Nhe*I are located in between the P*_spank_* promoter and the ribosomal binding site (RBS, yellow triangle) of the *gfp* gene, and permit the insertion of a sequence of interest in front of the *gfp* reporter gene.

**Figure 3 microorganisms-10-00587-f003:**
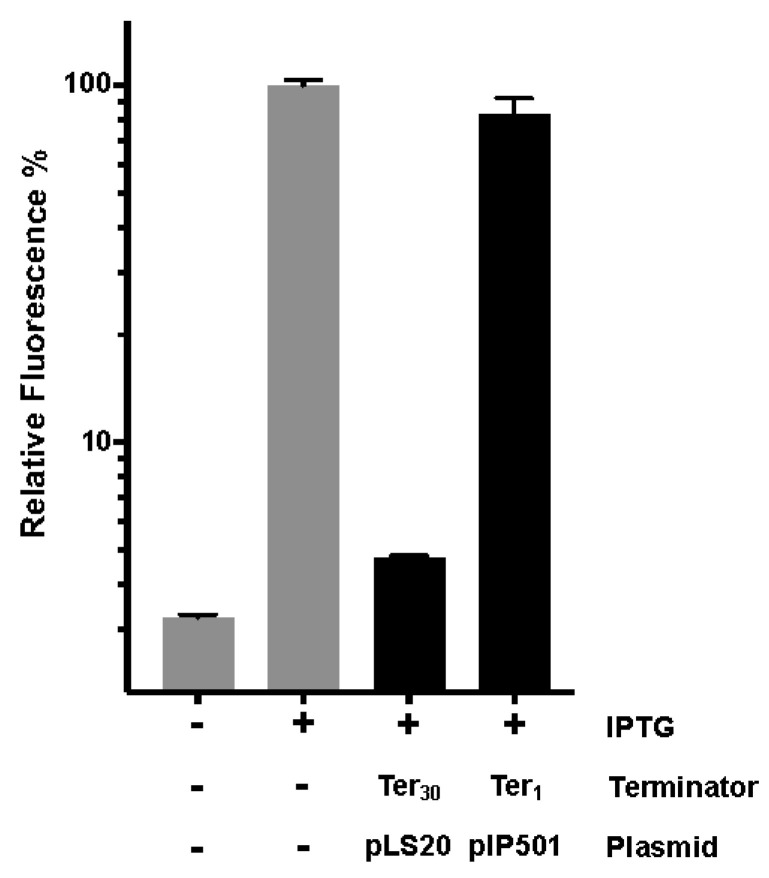
Fluorescence levels determined by flow cytometry analysis of cells grown in the absence or presence of 1 mM IPTG. Samples were withdrawn from late exponentially growing cultures (OD_600_ between 0.8 and 1). Gray bar: control strain AND101 (P*_spank_*-*gfp*). Black bars: strains AND127 (P*_spank_*-[Ter*_30_*]-*gfp*), indicated as Ter*_30_*, pLS20, and EEF7 (P*_spank_*-[Ter1]-*gfp*), right bar indicated as Ter_1_, pIP501. Fluorescence levels are expressed as the mean value of the Geomean values of 100,000 cells obtained in three independent experiments. Error bars represent standard deviation.

**Figure 4 microorganisms-10-00587-f004:**
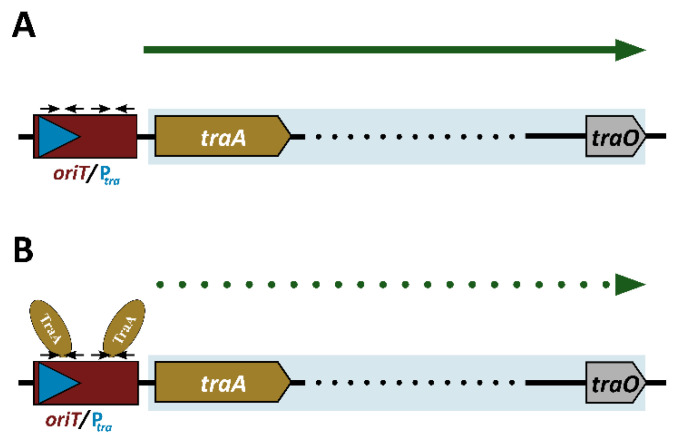
Regulation of the pIP501 main conjugation promoter P*_tra_* by the TraA relaxase. Schematic views of the activity of promoter P*_tra_* without (**A**) and with (**B**) relaxase-mediated repression. The TraA relaxase binds to two inverted repeat sequences (black horizontal arrows), overlapping with the P*_tra_* promoter region, thereby controlling the expression of the conjugation operon (genes *traA* to *traO*), as indicated by the green dotted line.

**Table 1 microorganisms-10-00587-t001:** Strains, plasmids and oligonucleotides used.

Strains	Description and Genotype	Source or Reference
*Escherichia coli*		
XL1-Blue	Used for regular cloning. *end*A1 *gyr*A96 (nal^R^) *thi*-1 *rec*A1 *rel*A1 *lac gln*V44 F’[Tn*10 pro*AB^+^ *lac*I^q^ Δ(*lac*Z)M15] *hsd*R17 (r_K_^-^ m_K_^+^)	Laboratory stock (Stratagene)
*Bacillus subtilis*		
168 (1A700)	*trpC2*	BGSC ^1^
AND101	*B. subtilis* 168 transformed with pAND101. *trpC2*, *amy*E:P*_spank_*-*gfp* (Spec^R^)	[30]
AND127	*B. subtilis* 168 transformed with pAND101. *trpC2*, *amy*E:P*_spank_*-[Ter_30_pLS20]-*gfp* (Spec^R^)	[30]
EEF7	*B. subtilis* 168 transformed with pEEF7. *trpC2*, *amy*E:P*_spank_*-[Ter_1_pIP501]-*gfp* (Spec^R^)	This work
**Plasmids**	**Description**	**Source or Reference**
pEEF7	pAND128 derivative containing terminator Ter_1_. Cloned fragment was generated by hybridization of primers oTer501_1A (*Sal*I) and oTer501_1B (*Nhe*I). (Amp^R^) and (Spec^R^).	This work
**Oligonucleotides**	**Sequence**	**Description**
oTer501_1A	**tcga**cGTAACGTCTGTTTATGCAGATGAATTTCACTTTTTATTGAAG	Hybridization primer for generating Ter_Inc18_ fragment. Used in combination with oTer501_1B. *Sal*I restriction site extension at the 5´end
oTer501_1B	**ctag**CTTCAATAAAAAGTGAAATTCATCTGCATAAACAGACGTTAC**g**	Hybridization primer for generating Ter_Inc18_ fragment. Used in combination with oTer501_1A. *Nhe*I restriction site extension at the 5’ end
pDR111_U_sec	TGACTTTATCTACAAGGTGTGGC	Forward primer for verifying sequences of PCR fragments cloned into pAND101.

^1^ Bacillus Genetic Stock Center, Biological Sciences 556, 484 W. 12th Ave. Columbus, OH, USA. 5´ nucleotides shown in bold, and lowercase correspondto the protruding overhangs generated after hybridization of the pair of complementary oligonucleotides. They are compatible with the overhangs generated by *Nhe*I and *Sal*I digestions.

**Table 2 microorganisms-10-00587-t002:** Putative terminators detected in the conjugation operons of Inc18 plasmids pIP501, pAMβ1 and pRE25.

Name	Plasmid	Position *	Sequence (5′-3′) ^†^
Ter1	pIP501	5076	GCAAATTGTAACGTCTGTTTATGCAGATGaaTTTCACTTTTTA
	pAMβ1	5076	GCAAATTGTAACGTCTGTTTATGCAGATGaaTTTCACTTTTTA
	pRE25	5076	GCAAATTGTAACGTCTGTTTATGCAGATGaaTTTCACTTTTTA
Ter2	pIP501	14.848	GTATTTATAAAAGCATGGTCGCAAGTTTCACTAGCAGCCATGCTTTTATTGAATC
	pAMβ1	14.853	GTATTTTTAAAAGCATGGTCGCAAGTTTCACTAGCAGCCATGCTTTTTTTGAATC
	pRE25	14.792	GTAAATTTAAAAGCATGGTCGCAAGTTTCACTAGCAGCCATGCTTTTTTTGAATC

* Positions correspond to the sequences shown in Appendix A. ^†^ Sequences predicted to form stem and loop when transcribed into RNA are shown in blue and red, respectively. Nucleotides in the sequence trailing the inverted repeat sequence that are not Ts are shown in lowercase.

## Data Availability

All data are included in the paper.

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
