# Peer review of "Conjugation Operons in Gram-Positive Bacteria with and without Antitermination Systems"

_microorganisms, 2022, doi:10.3390/microorganisms10030587_

Round 1
Reviewer 1 Report
Miguel-Arribas et al. submitted their study termed “Conjugation operons in Gram-positive bacteria with and with-2 out antitermination systems” to Microorganisms for publication. They reported here that plasmid pIP501, belonging to the Inc18 family of broad host-range conjugative plasmids, does not contain a processive antitermination (P-AT) system. This suggests that besides advantages, the presence of a P-AT system also attributes disadvantages. Possible (dis)advantages of a P-AT system and the implications for host range are discussed.
Major concerns:
- Abstract is not well wrote, we could not get key findings of this study.
- In section 3.1, there are too many information cited. If possible, these are should be described in the introduction. Results should not be like discussion.
- In general, this work is only the description of a biological event, but no mechanism, I think the authors at least should work out a mechanism involved in dysfunction of the P-AT system in pIP501.
Minor concerns:
- After the appearance of the first time, the generic name in a scientific name of one species should be abbreviated such as Escherichia coli in line 123, should by coli.
- Line 156-158, the sentence is unclear. Who was overnight cultured?
- The precise solution of PBS and pH should be given.
- Line 218, B. subtilis should be italic.
- There are many typos and errors in the text, please revise them carefully.
Author Response
Rebuttal to reviewer 1
Reviewer 1.
Miguel-Arribas et al. submitted their study termed “Conjugation operons in Gram-positive bacteria with and with-2 out antitermination systems” to Microorganisms for publication. They reported here that plasmid pIP501, belonging to the Inc18 family of broad host-range conjugative plasmids, does not contain a processive antitermination (P-AT) system. This suggests that besides advantages, the presence of a P-AT system also attributes disadvantages. Possible (dis)advantages of a P-AT system and the implications for host range are discussed.
Major concerns:
- Abstract is not well wrote, we could not get key findings of this study.
- In section 3.1, there are too many information cited. If possible, these are should be described in the introduction. Results should not be like discussion.
- In general, this work is only the description of a biological event, but no mechanism, I think the authors at least should work out a mechanism involved in dysfunction of the P-AT system in pIP501.
Minor concerns:
- After the appearance of the first time, the generic name in a scientific name of one species should be abbreviated such as Escherichia coli in line 123, should by coli.
- Line 156-158, the sentence is unclear. Who was overnight cultured?
- The precise solution of PBS and pH should be given.
- Line 218, B. subtilis should be italic.
- There are many typos and errors in the text, please revise them carefully.
Comments to reviewer 1
We thank the reviewer for her/his time and efforts to evaluate our manuscript.
The reviewer mentions that our abstract is not written well, because they could not get the key finding of this study.
We have re-phrased and shortened some of the sentences to make them easier to read. The key finding of our study is mentioned at the end of the abstract, which has now been modified slightly “…… some conjugation operons on ………… plasmids do not possess an antitermination system, suggesting that the absence of an antitermination system may have advantages”.
The reviewer mentions that section 3.1 contains too much information and that some of it should be placed in the Introduction. We agree with this comment. We have adapted paragraph 3.1 and moved some of the descriptive information to the Introduction.
The reviewer comments that our work describes a biological event but no mechanism, and s/he suggests working out a mechanism involved in the dysfunction of the P-AT system in pIP501. We would like to comment the following regarding this remark. As mentioned by the reviewer in her/his introduction of the reviewer report, we “reported here that plasmid pIP501, belonging to the Inc18 family of broad host-range conjugative plasmids, does not contain a processive antitermination (P-AT) system”.
Based on her/his statement here, it seems that the reviewer has misunderstood that plasmid pIP501 contains a P-AT system that is dysfunctional, which is incorrect. In the introduction and discussion of the paper, we describe that many conjugative operons present on plasmids harboured by Gram-positive bacteria contain a two-component P-AT system and multiple terminators. The presence of a conAn P-AT system provides at least two advantages: stricter control of expression of the conjugation genes by minimizing negative effects of spurious transcription and allowing differential expression of subsets of conjugation genes. Based on this, it seemed plausible that all conjugative operons present on Gram+ plasmids would be equipped with a P-AT system. We show that this is not the case for pIP501 and probably pAMβ1 and pRE25. Firstly, no sequences sharing similarity to P-AT systems or other antitermination systems could be identified on the conjugation operons of these plasmids. Secondly, for proper functioning a P-AT system is (almost) always located at or near the start of a (conjugation) operon, as shown for all the pLS20-related plasmids and all the hundreds of other plasmids containing a conAn type P-AT system. But, the pIP501 operon starts with the relaxase gene. Thirdly, the only putative terminator identified in the conjugation operons of these plasmids is not functional and lacks the main characteristics of a terminator. These analyses compellingly show that pIP501, and probably pAMβ1 and pRE25, do not contain a P-AT system.
We have previously shown that, for a P-AT system to operate, its component(s) must functionally interact with the elongation complex. We have also shown that the components of different conAn P-AT systems have different host range specificities, demonstrating that the P-AT system directly affects the distribution potential of the plasmid. Thus, whereas the presence of a P-AT system has advantages, it limits the conjugative host range of the plasmid.
The absence of a P-AT system will affect the flexibility and stringency of transcriptional control, but it relaxes conjugative host range limitations. The observations that pLS20 and pIP501 have narrow and broad conjugative host-ranges, respectively, are in line with these views.
Minor concerns:
- After the appearance of the first time, the generic name in a scientific name of one species should be abbreviated such as Escherichia coli in line 123, should by coli.
Thank you. Corrected at this position and corrections made throughout the paper.
- Line 156-158, the sentence is unclear. Who was overnight cultured?
Corrected. The revised sentence now reads. “In short, overnight grown cultures (37°C) of cells containing a transcriptional gfp fusion were diluted 100-fold in prewarmed fresh LB medium and grown at 37°C with shaking (180 rpm) until the culture reached an OD600 of 0.8-1.”
- The precise solution of PBS and pH should be given.
Done
- Line 218, B. subtilis should be italic.
Done, checked throughout
- There are many typos and errors in the text, please revise them carefully.
Corrected.
Reviewer 2 Report
This manuscript looks at whether a recently discovered antitermination mechanism, conAn, is present on a broad host-range Gram+ conjugative plasmid, pIP501 from the Inc18 family. This antitermination mechanism is widespread in other Gram+ conjugative elements. The study couples one computationally predicted terminator to GFP expression and finds that this putative terminator is not functional. The authors then explain that an antitermination system is probably not present in this conjugative element because they are often the first gene of the operon, and in this case that is a relaxase with known function. The authors conclude that all conjugative operons from the Inc18 plasmids do not contain functional intrinsic terminators, and that there is no evidence of anititermination. The authors then explain this absence may be due to the smaller size of these conjugation operons (spurious transcription is less of a problem in smaller operons), and that conAn type antitermination systems may pose limits on host range on their corresponding conjugative elements.
In my opinion, this manuscript draws conclusions that are not supported by its results. In essence, all the authors did was test one computationally predicted terminator on one conjugative element in one plasmid within the Inc18 family and conclude that ALL conjugative operons in this family do not use antiterminaton. Really, all they proved was that Ter1 within pIP501 is not a functional terminator. The authors imply that because there is high DNA sequence similarity between the conjugative elements within the Inc18 family (97-98%) their findings extend to the entire family of plasmids, but the terminator sequences are very small and could vary. This would be easy to check by comparing the sequence of predicted terminators between the three plasmids. Also, it’s not clear why the authors needed to test the terminator in the first place, if they found no evidence of proteins or RNA that could be from conAn or a similar anti-termination system in the first place. The conclusions of this manuscript need to be presented in a more cautious way, or additional work needs to be presented to extend these findings to the entire Inc18 family.
Minor comments:
Lines 36-52: The first paragraph of the introduction includes no references at all. References need to be added to support this section.
Lines 184-185: This sentence doesn’t make 100% sense. I guess you mean that you could’ve chosen any of the three plasmids (pAMB1, pRE25, or pIP501) because the DNA sequence identity is so high, but that doesn’t explain why you specifically picked pIP501.
Line 277: All analyses referenced in the results should be shown.
Caption Figure 1: The authors say the function of the following genes are known, and then sometimes put that function in parenthesis, for example traA (relaxase), but then sometimes just put something that does not explain the function in parenthesis, for example traE (virB4). A bit confusing.
Author Response
Rebuttal to reviewer 2
This manuscript looks at whether a recently discovered antitermination mechanism, conAn, is present on a broad host-range Gram+ conjugative plasmid, pIP501 from the Inc18 family. This antitermination mechanism is widespread in other Gram+ conjugative elements. The study couples one computationally predicted terminator to GFP expression and finds that this putative terminator is not functional. The authors then explain that an antitermination system is probably not present in this conjugative element because they are often the first gene of the operon, and in this case that is a relaxase with known function. The authors conclude that all conjugative operons from the Inc18 plasmids do not contain functional intrinsic terminators, and that there is no evidence of anititermination. The authors then explain this absence may be due to the smaller size of these conjugation operons (spurious transcription is less of a problem in smaller operons), and that conAn type antitermination systems may pose limits on host range on their corresponding conjugative elements.
In my opinion, this manuscript draws conclusions that are not supported by its results. In essence, all the authors did was test one computationally predicted terminator on one conjugative element in one plasmid within the Inc18 family and conclude that ALL conjugative operons in this family do not use antiterminaton. Really, all they proved was that Ter1 within pIP501 is not a functional terminator. The authors imply that because there is high DNA sequence similarity between the conjugative elements within the Inc18 family (97-98%) their findings extend to the entire family of plasmids, but the terminator sequences are very small and could vary. This would be easy to check by comparing the sequence of predicted terminators between the three plasmids. Also, it’s not clear why the authors needed to test the terminator in the first place, if they found no evidence of proteins or RNA that could be from conAn or a similar anti-termination system in the first place. The conclusions of this manuscript need to be presented in a more cautious way, or additional work needs to be presented to extend these findings to the entire Inc18 family.
Minor comments:
Lines 36-52: The first paragraph of the introduction includes no references at all. References need to be added to support this section.
Lines 184-185: This sentence doesn’t make 100% sense. I guess you mean that you could’ve chosen any of the three plasmids (pAMB1, pRE25, or pIP501) because the DNA sequence identity is so high, but that doesn’t explain why you specifically picked pIP501.
Line 277: All analyses referenced in the results should be shown.
Caption Figure 1: The authors say the function of the following genes are known, and then sometimes put that function in parenthesis, for example traA (relaxase), but then sometimes just put something that does not explain the function in parenthesis, for example traE (virB4). A bit confusing.
Comments to reviewer 2
We thank the reviewer for her/his time and efforts in evaluating our manuscript and providing suggestions that helped us to improve the quality of the manuscript.
The main critic of the reviewer is that according to her/him we have tested one computationally predicted terminator present on one conjugative element of one plasmid, and then conclude that ALL conjugative operons in this family do not use antitermination.
We agree in large part with this critique and have adapted our manuscript to be more precise and avoid over interpretation of our results. Thus, we have now specified in the introduction that the Inc18 group contains several plasmids whose DNA replication and segregation modules share high levels of similarity at the DNA level (>92%). Only a subgroup of these plasmids contains a conjugation operon and is self-transmissible. pIP501, pAMβ1 and pRE25 are the best-studied representatives of this subgroup.
The reviewer interpreted correctly that we have only tested Ter1 of pIP501. We should have explained that we did analyse all the three conjugation operons (pIP501, pAMβ1 and pRE25), and that the three Ter1 sequences are identical, and their locations with respect to the traE gene are also identical. To clarify this, we have now added the following information in the results section. First, we specify that we have compared the DNA sequences of the conjugation operons of pIP501, pAMβ1 and pRE25, and have presented an alignment showing that these sequences are highly similar as a supplemental figure. Second, we stipulate that we have analysed all three conjugation operons for the presence of putative terminators and include the results of this analysis in the revised manuscript (Table 2). Third, we describe that for all three conjugation operons, two putative terminators were found which we named Ter1 and Ter2, and that the three Ter1 sequences are identical and that the terminator is located at identical position in respect to the traE genes. We also describe that the three Ter2 sequences are located immediately downstream of the last gene of the conjugation operons (traO), and that these sequences are identical except for one position in one of the three sequences in the trailing T-stretch. Fourth, we also mention that we have screened the sequences of the three conjugation operons to identify possible intrinsic terminators that were not detected by the algorithms developed for identifying intrinsic terminators and focusing particularly on regions where the sequences are not fully conserved. This analysis did not reveal any other possible transcriptional terminators. Thus, as suggested by the reviewer, we have compared the sequences of the predicted terminators in all the three plasmids, and we show that the putative Ter1 terminator present in all the three plasmids (pIP501, pAMβ1 and pRE25) is not functional. We have also modified the text of the introduction, results and discussion sections and toned down our conclusions to avoid over interpretation.
Minor comments:
Lines 36-52: The first paragraph of the introduction includes no references at all. References need to be added to support this section.
Done. We have added eight references at different positions in this section to support the descriptions.
Lines 184-185: This sentence doesn’t make 100% sense. I guess you mean that you could’ve chosen any of the three plasmids (pAMB1, pRE25, or pIP501) because the DNA sequence identity is so high, but that doesn’t explain why you specifically picked pIP501.
In the revised version, we have explained in detail that the sequences of the conjugation operons of pIP501, pAMβ1 and pRE25 are highly similar, and that the positions and sequences of the putative terminator Ter1 are identical in all three plasmids. Thus, we have tested Ter1 that is present in all three plasmids.
Line 277: All analyses referenced in the results should be shown.
We have performed BlastN and BlastX searches of the three conjugation operons and did not find significant hits scoring with conAn or other antitermination systems. In our opinion, addition of the negative results of the blast searches does not improve the quality of the manuscript or contribute otherwise to a better understanding of the issue, which is why we have not included this.
Caption Figure 1: The authors say the function of the following genes are known, and then sometimes put that function in parenthesis, for example traA (relaxase), but then sometimes just put something that does not explain the function in parenthesis, for example traE (virB4). A bit confusing.
We have corrected this in the revised version and placed functionalities, rather than vir-homology, of the encoded proteins.
Reviewer 3 Report
accepts without any corrections
Author Response
Rebuttal to reviewer 3
Reviewer 3
We thank the reviewer for her/his efforts on evaluating our manuscript and the positive feedback.
Rebuttal to reviewer 3
Reviewer 3
We thank the reviewer for her/his efforts on evaluating our manuscript and the positive feedback.
Round 2
Reviewer 1 Report
At present, I don't think the quality of this submission is good eough to be published in this journal.
Author Response
In the first round of revision, this reviewer mentioned three major and five minor concerns. In the revised version of the manuscript, we addressed all eight concerns. Thus, we accepted the critique for seven out of the eight concerns and have adapted the manuscript as suggested by the reviewer. We also explained why we were not able to fully address the remaining issue, major concern three, which reads “In general, this work is only the description of a biological event, but no mechanism, I think the authors at least should work out a mechanism involved in dysfunction of the P-AT system in pIP501.” As outlined in our manuscript and explained in detail in the rebuttal, we provide compelling evidence that plasmid pIP501 does not contain a conAn P-AT system nor any other P-AT system.
The reviewer did not provide any additional comments or suggestions for further improvement of the revised manuscript. As we have addressed all issues raised by the reviewer in the first round of revision and no additional issues were raised in the second round, we feel we have adequately answered the concerns of the reviewer.